# A Multiple-Swarm Particle Swarm Optimisation Scheme for Tracing Packets Back to the Attack Sources of Botnet †

**Hsiao-Chung Lin**, **Ping Wang \***, **Wen-Hui Lin** and **Yu-Hsiang Huang**

Faculty of Department of Information Management, Kun Shan University, Tainan City, 710 Taiwan;
fordlin@mail.ksu.edu.tw (H.-C.L.); linwh@mail.ksu.edu.tw (W.-H.L.); s104001757@g.ksu.edu.tw (Y.-H.H.)

**\*** Correspondence: pingwang@mail.ksu.edu.tw; Tel.: +886-6-205-0545

† This paper is an extended version of our paper published in 6th International Conference on Applied System Innovation 2020 (IEEE ICASI 2020), Taitung, 5–8 November 2020.

**Abstract:** Network intrusion detection systems that employ existing IP traceback (IPTBK) algorithms are generally unable to trace multiple attack sources. In these systems, the sampling mechanism only screens parts of the routing information, which leads to the tracing of the neighbour of the attack source and fails to identify the attack source. Theoretically, the multimodal optimisation problem cannot be solved for all of its multiple solutions using the traditional particle swarm optimisation (PSO) algorithm. The present study focuses on the use of multiple-swarm PSO (MSPSO) for recursively tracing attack paths back to a botnet's multiple attack sources using the subgroup strategy. Specifically, the fitness of each path was calculated using a quasi-Newton gradient descent method to confirm the crucial path for successfully tracing the attack source. For multimodal optimisation problems, the MSPSO algorithm achieves an effective balance between individual particle exploitation and multiswarm exploration when premature convergence occurs. Thus, this algorithm accurately traces multiple attack sources. To verify the effectiveness of identifying Distributed Denial-Of-Service (DDoS) control centres, networks with various topology sizes (32–64 nodes) were simulated using ns-3 with the Boston University Representative Internet Topology Generator. The proposed A* search algorithm (minimal cost pathfinding algorithm) and MSPSO were used to identify the sources of simulated DDoS attacks. Compared with commonly available systems, the MSPSO algorithm performs better in multimodal optimisation problems, improves the accuracy of traceability analysis and reduces false responses for IPTBK problems.

**Keywords:** botnet; distributed denial of service; particle swarm optimisation; multiple-swarm PSO; IP traceback

## 1. Introduction

An increasing number of ready-made malware packages with graphical user interfaces allow users with limited computer skills to conduct complex network attacks and compromise targeted computers through hostile actions, including opening ports, injecting program instructions through encrypted channels and executing these malicious programs. Hackers then perform cyber-attacks on the compromised computers. There are common types of cybersecurity attacks, such as distributed denial-of-service (DDoS) attack, phishing attack, email spam, etc. In the case of web service attacks, the herder tends to terminate the web services of opponents through DDoS attacks or the theft of business information. The aforementioned convenient attack tools have intensified the already severe challenges of network security. In a DDoS attack, a malicious user, who is referred to as a herder, utilises a Command and Control (C&C) server to initiate a botnet attack on a single target or multiple systems. The resulting flood of incoming messages overwhelms the target and causes it to shut down and deny access to legitimate users [1]. Typically, DDoS attacks can be divided into two major categories: Bandwidth attacks and resource attacks. A bandwidth attack simply attempts to generate packets to flood the victim's network so that

legitimate requests cannot be sent to the victim's machine. A resource attack aims to send packets that misuse network protocols or malformed packets that tie up network resources so that resources are not available to legitimate users [2].

A botnet is a group of computers that is compromised by malware sent from and manipulated by a herder. After malware has been successfully installed in a victimised system, the victimised system becomes a zombie (compromised host) and will accept remote commands from the herder. Typically, botnet C&Cs play a vital role in the operations conducted by cybercriminals who use infected machines to send spam, send ransomware, launch DDoS attacks, commit e-banking fraud, commit click fraud or mine cryptocurrencies, such as Bitcoin. According to statistical reports from Spamhaus Malware Labs, systems identified and blocked 17,602 botnet C&C servers hosted on 1210 different networks worldwide in 2019. The 2019 figures represent a 71.5% increase from the number of botnet C&Cs observed in 2018 [3].

The problem of identifying the origin of an attack over the internet to enable the flooding controls for web service servers is referred to as the IP traceback (IPTBK) problem. A typical botnet uses one to many control links between its C&C server and its victims. Two main types of bots are used in DDoS attacks: IRC and peer-to-peer bots. As illustrated in Figure 1, attack path reconstruction is a typical IPTBK technique used against IRC bot attacks. As displayed in Figure 1, attackers use zombies controlled through their C&C server to generate a high volume of packets for flooding the victim's network over the internet so that legitimate requests cannot access the service.

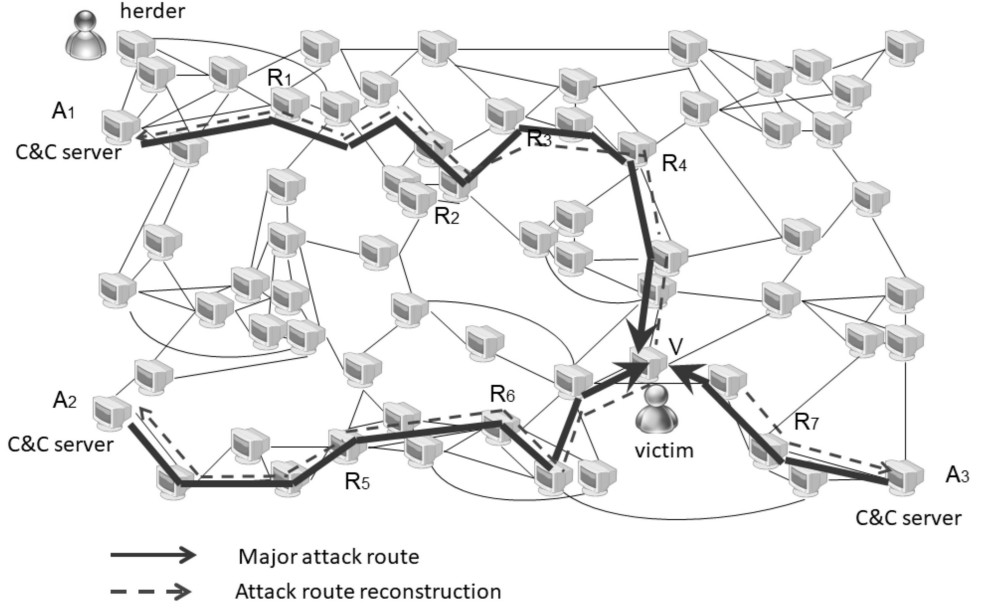

**Figure 1.** Attack path reconstruction for the IP traceback (IPTBK) problem against IRC bot attacks.

In practice, solving the IPTBK problem entails a trade-off between the number of routing packets collected and the accuracy of the traceback. Typically, the solution to the aforementioned problem involves the following crucial steps: (i) Collecting sufficient route information to determine all the possible routes between the attacker and the victim for given constraints on the quantity of routing packets collected and the computational time and (ii) identifying the minimum size of the evidence for identifying more than 95% of the attack sources.

The defence of network services against DDoS attacks typically involves the use of a combination of traffic classification, attack source detection and threat response. Defenders seek to block traffic that they identify as illegitimate and to allow traffic that they identify as legitimate. In Figure 1, the route was reconstructed using the IPTBK scheme for rebuilding

attack paths between the victim and attackers (e.g., route V-$R_6$-$R_5$-$A_2$). The possible paths of DDoS attacks were calculated from the relative probability of each reachable route.

Problem definition: Typically, the IPTBK problem involves collecting sufficient route information to determine all the possible routes between the attackers and the victim for given constraints on the quantity of routing packets collected and the computational time [4]. To defend against DDoS attacks from botnets, defenders often used ingress filtering [5], probabilistic packet marking (PPM) [6–8], ICMP trackback messages (itrace) [9] or deterministic packet marking (DPM) [10] approaches for sampling and sorting the passing routing packets. These IP traceback techniques are summarized in Table 1.

**Table 1.** Techniques for IP traceback.

| | Features | | Limitations |
|---|---|---|---|
| | **Data Collection** | **Route Reconstruction** | |
| ingress filtering [5] | Each router marks all traversing packets for information flows (heavy loading) | • The attack path reconstruction is based on collected packets. <br> • Route reconstruction process is started from victim via upstream links and recursively repeat until the true of attack source is located. | When IP addresses are used through a proxy or a spoofed IP address, which does not identify a specific user within that pool of users. |
| PPM [6–8] | Each router marks packets with some probability $p$ for information flows to reduce the consumption of computational resources | • The attack path reconstruction is based on collected packet digests. <br> • Route reconstruction process is same as that of ingress filtering. | • Loss of marking information. <br> • The difficulties to reconstruct attack path□associated with low accuracy. |
| iTrace * [9] | When forwarding packets, routers can generate a Traceback message based on icmp command that is sent along to the destination with a low probability. With enough traceback messages from enough routers along the path, the traffic source and path can be determined. | The attack path reconstruction is based on traceback messages, which recursively repeat until the true of attack source is located based on traversing IPs. | Multiple attack sources will generate multiple attack paths that make traceback more difficult to locate the original attack sources. |
| DPM [10] | Using an advanced marking mechanism with ingress edge routers and traversing routers. | The same as that of PPM | DPM is sensitive to router subversion because it is critical that marks, once placed in packets, are not over written. |

(*) Basically, iTrace frequently does not work because the icmp command is inhibited by servers and managers to protect from malicious scans.

They then use optimised route-searching algorithms to reconstruct the attack paths and trace the attack sources to determine (1) the lower bound of the size of the routing packet to achieve a certain level of traceback accuracy, (2) how to trade-off between the number of attack packet used for traceback vs. the accuracy of the traceback and (3) the minimum size of the routing packet for identifying more than 95% of the attack sources. Existing IPTBK research assumes that complete attack path packets can be collected and traced. However, in practice, practical PPM and DPM mechanisms can only sample and collect small fractions of network routing information (e.g., 3%). Accurately tracing the C&C of a botnet is an information security challenge.

The IPTBK problem is an NP-hard problem in which the aim is to attain a set of feasible solutions within polynomial time under certain constraints (e.g., a limitation on the number of nodes within the topology). To reduce the computational load of reconstructing the attack path, selecting an effective traceability algorithm with low complexity is a key

consideration. Optimised route-searching algorithms, including the genetic algorithm (GA), particle swarm optimisation (PSO) [11–14], artificial neural networks, simulated annealing, tabu search, evolutionary algorithms and ant colony optimisation algorithm (ACO), are commonly used for reconstructing the attack path.

PSO is an artificial intelligence technique that can be used to solve a diverse array of combinational optimisation problems, including vehicle routing problems (VRPs) [15], wireless sensor network (WSN) problems [16] and network threats [17]. We found that PSO is a simple algorithm suitable for solving the IPTBK problem in future high-speed networks and that a variant of PSO outperforms existing optimised route-searching models.

The PSO has a high convergence speed for IPTBK but easily falls into local optimal solutions. Thus, particles mistakenly fly toward a local optimum without exploring other regions of the search space. Consequently, the algorithm is trapped in the local optimum and converges prematurely [14]. A more promising approach is to optimise the search for multiple attack sources with different subswarms of particles to enlarge the search space, i.e., multiswarm PSO (MSPSO). The MSPSO is a variant of particle swarm optimisation based on the use of multiple subswarms instead of one swarm. The general approach in multiswarm optimisation is that each subswarm focuses on a specific region while a specific diversification method decides where and when to launch the subswarms. The multiswarm design is especially fitted for the optimisation on multimodal problems, where multiple (local) optima exist. [18]

Therefore, we adopted the multi-subswarm strategy to expand the search scope of the particle swarm and increase the search efficiency in large networks. We developed a multiple-swarm PSO (MSPSO) scheme, which is designated as MSPSO-IPTBK, to analyse the performance improvement of IP traceability in the attack path reconstruction process for different dynamic network topologies. The validity of the proposed algorithm was demonstrated by performing a series of experiments using ns-3 simulator with the BRITE (the Boston University Representative Internet Topology Generator) topology generator. The ns-3 tool is free, licensed under the GNU GPLv2 license and publicly available for research, development and use in academic network simulations. The BRITE is a topology generation framework constructed with ns-3 to create large-scale internet topologies efficiently. It takes advantage of ns-3's simulation capabilities.

In summary, the primary contributions of this study are as follows:

- The multimodal optimisation problem was solved using the MSPSO method. Moreover, different botnet attack sources were analysed with high accuracy for the reconstructed attack path to identify the most probable attack paths. This identification was performed for assisting security managers to identify the sources of DDoS attacks.
- The optimal route-searching process of the MSPSO algorithm was improved to prevent the PSO algorithm from converging prematurely to a local suboptimal solution in a large search space.
- The accuracy of the MSPSO algorithm was obtained as 95.89% for the effects of dynamic traffic in the experimental network (number of nodes = 24) and as 94.64% for the network topology (number of nodes = 64).
- Compared to traceback accuracy with other route search algorithms such as the A* algorithm [19] and the PSO [11–14], the performance of the MSPSO-IPTBK algorithm in reconstructing attack paths provides superior performance for analysing the attack origins from multiple different data sources using ns-3 with the BRITE framework.

The remainder of this paper is organised as follows. Section 2 presents previously published studies in the IPTBK field and describes the use of existing methods in solving the IPTBK problem. Section 3 describes the MSPSO scheme for solving the IPTBK problem. Section 4 presents the experimental results obtained using the MSPSO algorithm and describes the global heuristic performance of the algorithm. Finally, Section 5 provides some concluding remarks.

## 2. Overview of PSO Schemes and the IPTBK Problem

This section reviews several existing methods for solving the IPTBK problem in DDoS attacks and introduces MSPSO schemes for identifying possible attack sources.

### 2.1. Existing IPTBK Techniques for DDoS Attacks

Typically, IPTBK techniques can be classified into two categories on the basis of when attack traffic flows are traced from the victim to the sources: (i) Ongoing and (ii) post-mortem traceback techniques. Ongoing traceback is a traceback method in which the source of hacking attacks is traced back in real time. In general, attack network flows must be specified from normal network flows immediately. In practice, the real-time tracing of the remote controller is extremely difficult because its true identity is generally disguised by constantly changing spoofed IP addresses. Typical ongoing traceback schemes include input debugging, overlay networks and link testing.

In post-mortem traceback techniques, the partial path information is filtered and stored for rebuilding the attack paths. This behaviour can assist managers in identifying the attack sources after an attack. Post-mortem traceback techniques include ingress filtering [5], Source Path Isolation Engine (SPIE) [5], PPM [6–8], iTrace [9] and DPM [10].

In PPM [6–8], each router marks packets with some probability *p* for information flows. For example, *p* = 1/1000 implies that 1 packet is marked for every 1000 packets received to reduce the consumption of computational resources during the forwarding process. In the traffic information collection process, PPM must record partial path information. Routers are required to modify the header of each packet to store the traffic information in the router, which implies that multiple attackers cannot be traced simultaneously. DPM [10] was proposed to overcome the disadvantages of PPM using an advanced marking mechanism with ingress edge routers and traversing routers. Any spoofed source address can impair DPM. To avoid tracing a counterfeit attack path from spoofed IP addresses, scholars have proposed SPIE, which is a hash-based technique for IPTBK that generates audit trails for traffic within suspicious networks. Researchers must use efficient schemes, such as PSO, ACO and the GA with the SPIE mechanism, to reduce the computational complexity of route reconstruction by filtering partial path information. They must also provide countermeasures against spoofed IP attacks.

### 2.2. MSPSO Algorithms

Route construction is performed using a velocity-state change rule for position updating based on two factors, namely $P_{best}$ and $P_{gbest}$, as illustrated in Figure 2. When a PSO system solves the IPTBK problem, each particle swarm constructs a route between the victim and the attacker by repeatedly applying velocity-state updating rules, which are expressed using Equations (1) and (2). As presented in Equation (1), the speed of each particle is updated by updating the velocity of each particle toward its optimal locations $P_{best}$ and $P_{gbest}$ at each time step in the PSO process. [14]

$$v_i(t+1) = w_i v_i(t) + c_1 \cdot rand() \cdot (P_{best} - x_i(t)) + c_2 \cdot rand() \cdot (Pgbest - x_i t)) \quad (1)$$

$$x_i(t+1) = x_i(t) + v_i(t+1) \quad (2)$$

$$w_i = \frac{2}{\left| \left( 2 - c - \sqrt{c^2 - 4c} \right) \right|} \quad (3)$$

$$c = c_1 + c_2, c \geq 4, \quad (4)$$

where *v* and *x* represent the speed and position of particle *i* (*i* = 1, … , *m*) at time *t*, respectively, and $c_1$ and $c_2$ represent the acceleration constants. Suitable acceleration constants ($c_1$ and $c_2$) can control the speed of the particle's movement. Typically, $c_1$ is equal to $c_2$, and both these parameters are equal to 2. The parameter *rand*() represents a random number in the range (0,1).

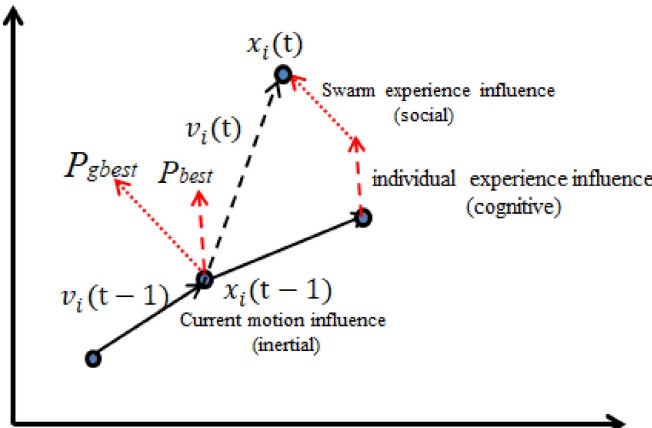

**Figure 2.** Attack path reconstruction using a particle swarm optimisation (PSO) scheme [13].

Notably, each particle utilises its memory to record its previous best position and the best position discovered by any particle in the swarm. Thus, when the particle moves to a new position, the fitness value of the new position is calculated. If the fitness value of the new position is better than that of the previous best position (i.e., $P_{best}$), then the value of $P_{best}$ must be replaced by that of the new position updated according to the particle's optimal experience. Similarly, $P_{gbest}$ must be replaced by $P_{best}$ if the fitness value of the new position is better than that of $P_{gbest}$.

In searching for the global optimal solution, the movement of each particle in the subswarm is affected by three factors: (1) Inertia, (2) individual experience and (3) swarm experience. [11,12].

To prevent the particle swarm from rapidly converging on a single path, the proposed MSPSO algorithm divides a particle swarm into several subswarms. In the proposed MSPSO algorithm, the local update rules in each subswarm are required to search on multimodal optimisation problems. The multimodal optimisation methods generally arrange the exploration and exploitation into different search stages as shown in Figure 3 [20]. In the local search stage, each subswarm is used to explore possible solutions. In this stage, the focus is on discovering the local optimal solutions in the region. In the global search stage, the global optimal position is exploited through a regrouping strategy.

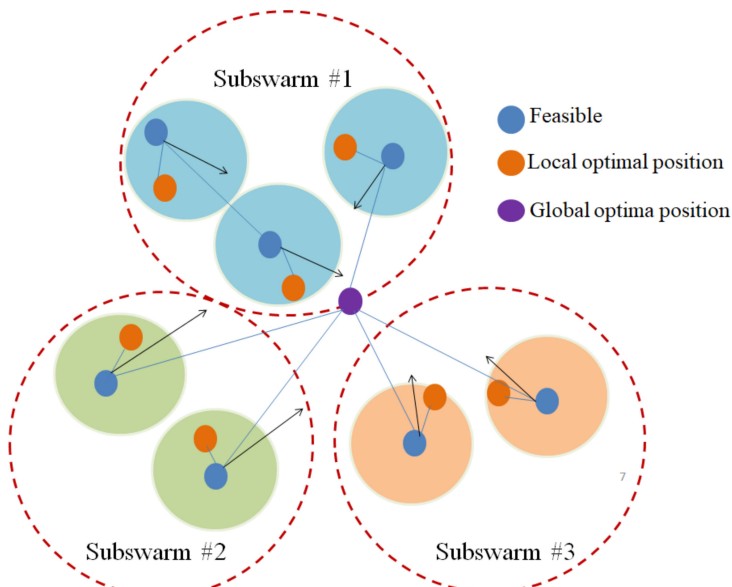

**Figure 3.** Dynamic behaviour of the particle movement for three multiswarms in multiswarm PSO (MSPSO) [20].

Two known multimodal optimisation methods use revised PSO schemes: The waves of swarm particles (WOSP) [21] and the dynamic multiswarm PSO (DMS-PSO) [22,23] algorithms. The two crucial algorithms are introduced as follows.

### 2.2.1. WOSP

When particles get too close, particles in WOSP approach are expelled by a short-range force into new waves (i.e., subswarms), thus avoiding a complete convergence. To perform the multiswarm exploration process when particles are settling close together, the WOSP scheme [21] introduces a short-range force (SRF) on the basis of particle collision. A short-range interaction between particles, that is, a gravitational-style attraction, produces a force of attraction between particle *i* and particle *j*. The magnitude of the force is inversely proportional to some power *p* of the distance between the particles. The SRF generates a velocity component $v_{ij}$ for the movement of one particle *p* toward the other. This component can be represented as follows:

$$v_{ij} = K/d_{ij}^{p}(t), \tag{5}$$

where $d_{ij}$ represents the distance between particles *i* and *j* and *K* is a constant.

### 2.2.2. DMS-PSO

To increase the diversity of particle swarm searching in multimodal optimisation problems, the DMS-PSO algorithm [22,23] uses a regrouping strategy in which the entire population is divided into a large number of subswarms. These subswarms are regrouped frequently using various regrouping schedules. Information is exchanged among the particles in the entire swarm. The DMS-PSO uses two phases to enhance the search performance:

(1) Local search phase: A quasi-Newton method (e.g., the Broyden–Fletcher–Goldfarb–Shanno (BFGS) algorithm) is used to speed up the search for the best position in a subswarm. Notably, the space constrained within the range is searched in this phase. The fitness value of a path is calculated by minimising the cost function of traversed paths.

(2) Convergence phase: After completing the local search process, the DMS-PSO algorithm periodically regroups the particles of the subswarms into new subswarms. The new subswarms initiate the search process again with the previous particle swarms until the global best position is found.

Compared with those of the MSPSO methods for IP traceback problems, we regrouped the particles of the subswarms, such as WOSP and DMS-PSO, which have produced pretty good results and are generally considered as the best solutions. Finally, we made the comparisons with those of MSPSO schemes, as shown in Table 2.

**Table 2.** Feature comparisons of three known MSPSO schemes.

| Scheme | Feature | Advantage | Limitation |
|--------|---------|-----------|------------|
| WOSP [21] | When particles get too close, they are expelled by a short range force into new subswarms, thus avoiding a complete convergence. | WOSP is especially fitted for the optimisation on multimodal problems using the SRF where multiple local optima exist. | Sometimes, it may generate loop iterations in the searching process to discover the optimal solution. |
| DMS-PSO [22,23] | DMS-PSO periodically regroups the particles of the subswarms after they have converged into new subswarms, and the new swarms are started with particles from previous swarms. | DMS-PSO can achieve a good balance between the exploration and exploitation abilities in multimodal problems. | DMS-PSO separates the optimal solution searching process into distinct phases, which could weaken both mechanisms of the search process. |

## 3. Application of the Proposed MSPSO Algorithm to the IPTBK Problem

The proposed MSPSO algorithm was used to analyse the accuracy of the reconstruction attack path for different topology sizes. The basic IPTBK problem is discussed in the following subsections.

The problem of attack path reconstruction can be expressed as a directed graph as $G = (X, E) = (x_i, e_i)$, where $X$ represents a set of nodes, $x_i = \{x_{i1}, \ldots, x_{id}, \ldots, x_{iD}\}$, $x_s$ is a set of source nodes (i.e., attack sources), $x_d$ is a set of sink nodes (i.e., victims) and E denotes the edge $e_{ij}$ of the graph from node $x_i$ to node $x_j$ in $D$-dimensional space.

### 3.1. Basic Idea

Assume that multiple feasible attack paths exist among the nodes $(v_i, \ldots, v_k, \ldots, v_j)$. The fitness value of each route was calculated to examine whether the particle selects a low-cost routing path. Typically, a path-search algorithm is used to enable travel routing systems to increase their efficiency by considering the selection of low-cost network routing, namely (i) the shorter distance between two nodes $(x_i, x_j)$ and $d_{ij}$(hop count) and (ii) the better quality of service (QoS) between nodes $(x_i, x_j)$ (i.e., QoS$_{ij}$). Thus, the path-search algorithm usually selects a specific path with the smallest routing cost (i.e., least distance travelled and high QoS to decrease the routing time). Typically, the routing cost of path $(C_i)$ for node $x_i$ to the victims is inversely proportional to the distance travelled and directly proportional to the QoS (i.e., low delay and traffic congestion), i.e., $C_i = f\left(QoS, \frac{1}{d_{ij}}\right)$. Theoretically, solving the problem of multimodal optimisation requires seeking the minimum cost function of $L_p$ subject to the constraint of routing cost. The minimum cost function of $L_p$ is a positive number $(\sum_{j=1}^{n} C_i \cdot e_{ij} > 0)$ that is expressed as follows:

$$Fitness = Lp = \sum_{j=1}^{n} C_i \cdot e_{ij}^x, \tag{6}$$

$$\begin{aligned} &Min\ Lp,\ \forall i, j \\ &Subject\ to\ \sum_{j=1}^{n} C_i \cdot e_{ij}^x > 0, \end{aligned} \tag{7}$$

where *Fitness* represents the fitness value of a path. The *Fitness* is used for assessing the suitability of each path, and $e_{ij}^x$ indicates whether a path exists from node $i$ to node $j$ for particle $x$. An $e_{ij}^x$ value of 1 indicates that a connected path exists from node $i$ to node $j$ for particle $x$, whereas an $e_{ij}^x$ value of 0 indicates that no connected path exists.

In the proposed MSPSO scheme, the path-search algorithm uses a best-first search strategy formulated using weighted graphs to discover the lowest-cost path to the given goal node. In general, it achieves this aim by examining a tree of paths originating at the start node and extending these paths one edge at a time until its termination criterion is satisfied.

Cluster-first, route-second strategy: Inspired by the capacity-constrained vehicle routing problem and its cluster-first, route-second solution (CFRS) strategy, this study divided the attack sources into several network areas based on IP domains and the time from DNS log data. Moreover, several particle subswarms were allocated in each area in sequence. One might expect to obtain the optimal solution in the entire domain with a heuristic algorithm. The advantage of the proposed strategy is that the attack sources are divided into several areas in advance by performing clustering analysis on routing traffic, which can reduce the redundancy of reconstructing the attack path.

### 3.2. Solving the IPTBK Problem Using the MSPSO Algorithm

In the proposed scheme, a PSO with subswarm searching based on CFRS strategy is used as a global optimisation algorithm for solving problems in which possible routes between the victim and the attack origin(s) can be represented as a multimodal optimisation problem. The detailed flowchart for the proposed model is illustrated in Figure 4. Figure 4 displays the three subphases in the attack path reconstruction process: (1) The data pre-processing phase, (2) route construction phase and (3) model validation phase.

For evaluating the effectiveness of IPTBK for the proposed model, the simulation tool ns-3 was used with the BRITE to stimulate a large network threatened by DDoS attacks. The problem definitions assume that the attacks are complicated by various tricks, such as indirect attacks and spoofed IP attacks. The victim uses Tcpdump to capture information on network traffic (in Pcap files) for further analyses.

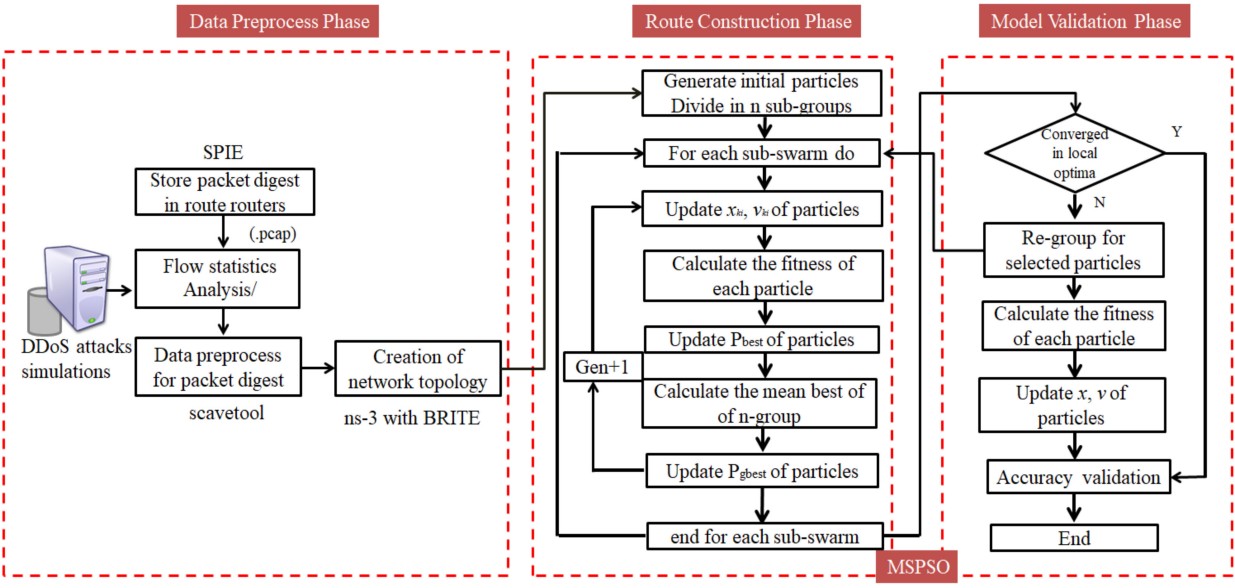

**Figure 4.** Basic concept of route construction with the revised PSO model.

### 3.2.1. Data Pre-Processing Phase
Data Collection

Assume that the IPTBK problem must be solved using the MSPSO algorithm with an SPIE involving a Bloom filter and an auditing function of packet marking, as displayed in Figure 5. In the solution procedure, the associated routers store partial route information during the attack process (i.e., record packet digests of sampled packets, $p$%). In the study, we used a small sampling rate $p$ (3.0%) to achieve low storage and computational cost for the networks. The victim must use recursive lookup to reconstruct the attack path associated SPIE mechanism. The SPIE uses auditing techniques to support the attack path construction from traffic flow packets collected from the true source of the network. Notably, traffic auditing is accomplished by computing and storing 32-bit packet digests rather than storing the packets themselves in the SPIE. Additional detailed information regarding the format of 32-bit packet digests in the IP header is provided in [5].

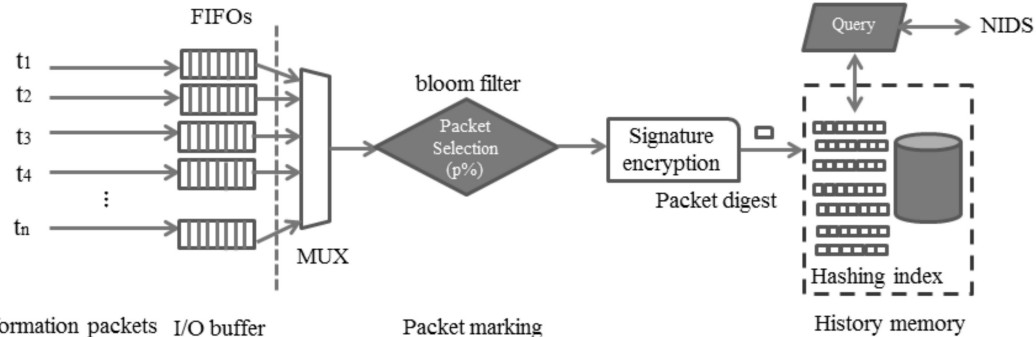

**Figure 5.** Packet marking mechanism with a Bloom filter and an auditing function in the Source Path Isolation Engine (SPIE) [5].

Encoding of Particles

To search each of the weighted graphs in terms of the routing cost, the network packets collected were encoded for examining each path, with the routing costs represented by weights. The system must list the connection sequence of all the routing paths and then select the appropriate path by comparing the costs of attack paths. An example of the format of particle encoding is presented in Figure 6. In Figure 6, $x^k_i$ represents the position and velocity information of a particle in subswarm $k$.

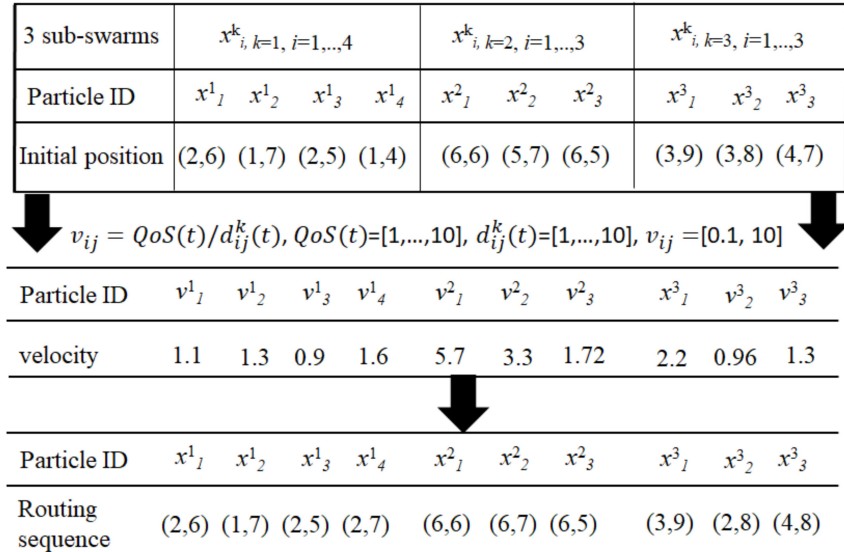

**Figure 6.** An example of the state update of the PSO particles in $k$ subswarm movement.

Intelligent Reconnaissance Strategy

The position and velocity vectors of the particles in the PSO method are represented by real numbers of the form $x^k_i$, $x^k_j$. The path-search algorithm searches each of the weighted graphs with appropriate routing costs. First, the sequence of each attack path is examined with a sigmoid function to decide whether two neighbouring nodes are connected. In Equation (6), $e_{ij}$ indicates whether the system has a path that connects node $i$ to node $j$. The parameter $e_{ij}$ is affected by the constants $c_i$ in Equation (6) if the $c_i$ *value* is higher than the minimum cost (threshold) to connect two nodes $t_c$. As displayed in Figure 7, this node is connected on the path. Finally, the attack path is formed by linking all the connected nodes with a list of IP addresses.

$$Sig\ value = \text{array}\left(\frac{1}{1 + e^{-(x - t_c)}}\right) \tag{8}$$

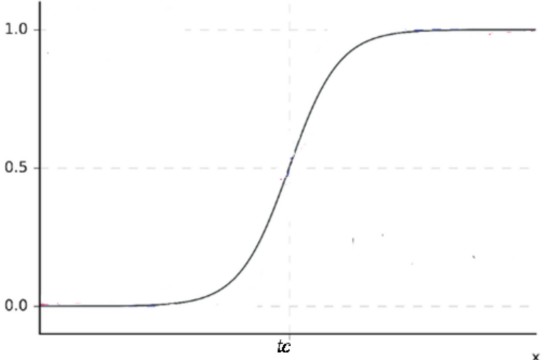

**Figure 7.** Sigmoid function.

### 3.2.2. Route Construction Phase

Because traditional PSO algorithms tend to route particle swarms to similar paths with the smallest cost in a large network topology, they cause rapid accumulations of particles to converge to a single path. The subgroup strategy [21] is expected to decrease the speed of accumulation of particles and increase the search space with high accuracy on multiple routing paths for identifying all the attack sources. As displayed in Figure 8, a two-group strategy was implemented in the attack path-search process.

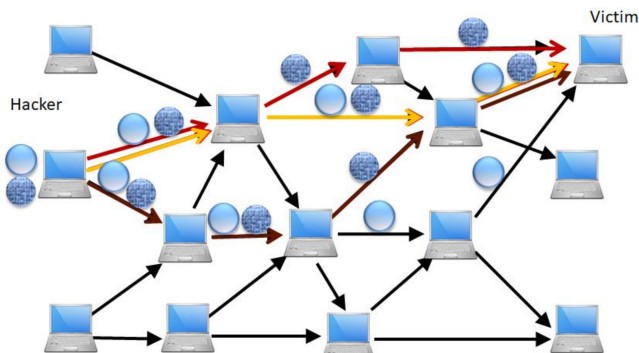

**Figure 8.** Improvements in the PSO algorithm when using the two-group strategy in the route-searching process.

For example, the original particle swarm was divided into two, three and five subswarms to compare the obtained convergence performance with that provided by a particle swarm in the traditional PSO algorithm. In addition to the local and global update of the position and velocity vectors, the fitness value of each path in each subswarm must be updated to evaluate whether the particles travel on attack paths back to the attack sources.

In the following, the MSPSO combines local search process with global search algorithms to perform the attack path reconstruction based on collected packet digests, which starts from victim via upstream links and recursively repeats until the true of attack source is located as follows.

Local Search Process

Inspired by DMS-PSO [22,23], the particle swarm uses a greedy local search technique associated with the quasi-Newton gradient descent method (BFGS) to identify the possible local optimal positions with an intelligent reconnaissance strategy, as presented in Equation (9). Theoretically, the BFGS method can efficiently search for the individual optimal position ($P_{\text{best}}$) and the optimal position in a subswarm ($P_{gest}$) in a convex space. Moreover, it can efficiently improve the solution quality of each particle. To determine $P_{\text{best}}$ and $P_{\text{gbest}}$ for a subswarm, BFGS can be used for dynamically adjusting the particle acceleration (weight $w_i$) for avoiding overfitting by minimising the routing cost $C_i$, which is expressed in Equation (6).

$$w_i(t+1) = w_i(t+1) - \eta \frac{\partial C_i}{\partial w_i}, \tag{9}$$

where $\eta$ is the learning factor.

The recursive process with the aforementioned updating rule causes the $P_{\text{best}}$ and $P_{\text{gbest}}$ to use an estimated fitness value for each particle. Moreover, DMS-PSO allows large amounts of information exchange in a subswarm to enhance the movement ability of some individual particles. A fraction (e.g., 30%) of the particle subswarm is selected to remain at the same value as $P_{\text{gbest}}$ in a subswarm to improve the local search results when the particle position is updated.

Once the $P_{\text{best}}$ and $P_{\text{gbest}}$ in a subswarm have been calculated, they must be used to update the velocity and position vectors. Considering the practical requirements and

limitations of traceback to the sources of DDoS attacks, this study revised the updating rule of the MSPSO algorithm to formulate the position and velocity update rules as follows:

$$v_i^k(t) = w_i \cdot v_i^k(t-1) + c_1 \cdot rand() \cdot \left(p_{\text{best}} - x_i^k(t-1)\right) + c_2 \cdot rand() \cdot \left(p_{\text{gbest}} - x_i^k(t-1)\right) \tag{10}$$

$$x_i^k(t) = x_i^k(t-1) + \Delta t_{ij}^k(t-1) \tag{11}$$

$$\Delta t_{ij}^k(t-1) = \begin{cases} v_i^k(t) \cdot QoS/d_{ij}^k & \textit{if the path is on the optimal path in } k - subswarm \\ 0 & \textit{otherwise} \end{cases} \tag{12}$$

In Equation (11), $\Delta t_{ij}^k(t-1)$ represents the movement distance of $\Delta t$ in Equation (10), which is inversely proportional to the path distance $d_{ij}^k$ of the 2 end nodes. The parameter $d_{ij}^k$ is the number of hops on the $i$ attack path in $k$ subswarm.

Global Search Process: Dynamic Neighbourhood

To mimic the sociological phenomenon where individuals indirectly share information with others located around themselves, the proposed MSPSO algorithm implements a reorganisation plan of the particle swarm when premature convergence occurs. Therefore, the particles are forced to exhibit a dynamic change in the neighbourhood structure when all the particle subswarms quickly converge on a single attack path. The reorganisation plan of the particle swarm regroups the particles from neighbouring subswarms to the new configuration, thereby expanding the search space of each subswarm (Figure 9). In every $R$ generation of each iteration, a subset of subswarms is randomly regrouped and uses the new configuration of the subswarms to search for neighbouring areas. The parameter $R$ is the regrouping period [22]. Through the aforementioned method, each subswarm can fully exchange information, which increases the diversity of the particle swarm search. The new neighbourhood structure has a higher degree of freedom than the traditional static neighbourhood structure does.

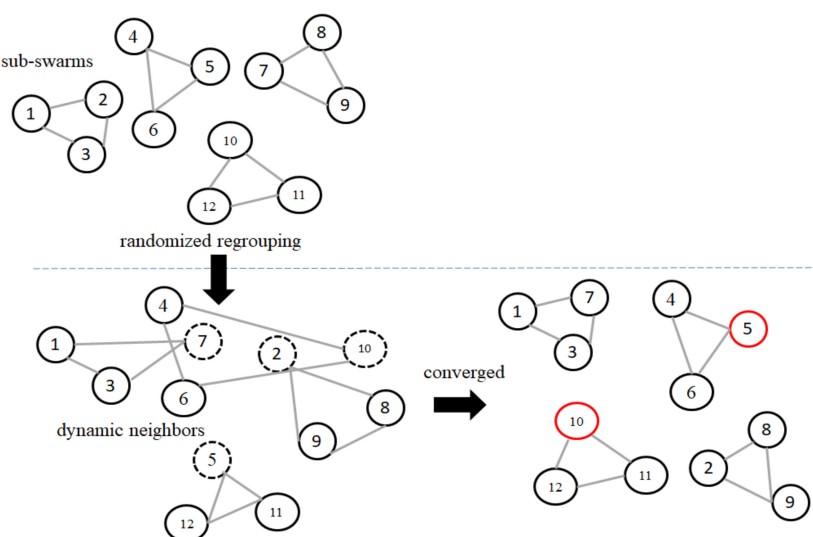

**Figure 9.** Dynamic regrouping for four particle subswarms.

3.2.3. Model Validation Phase

Once the position of the particle in the space search process has been sufficiently updated using Equations (9) and (10), the particle $p$ has traversed the most probable attack paths back to attack sources (the nest). All the network nodes visited by each particle swarm up to time $t$ are memorised in a tabu list to prevent the particle swarm from revisiting the same node and creating a circular path during the search process. The aforementioned process is repeated iteratively for a preset number of cycles. Finally, the particle swarms

know the appropriate route that leads them back to the nest (i.e., they have discovered the origin of the attack).

The model accuracy was evaluated using the coverage percentage (%), which is the ratio of the average number of packets on an attack path to the total number of routing. The coverage percentage is expressed as follows:

$$\text{Coverage percentage or precision (\%)} = \text{(Average number of packets/attack path)/(total number of routing packets)}, \qquad (13)$$

where the average number of packets on the attack path is computed as the total number of packets on the route divided by the routing distance (in terms of hop count). If the converged solution is not the true attack node, then the average number of packets on the route is reset to 0 and the search for the true route is resumed. The detailed algorithm for route construction with MSPSO-IPTBK model is presented in Figure 10.

While (the number of max_gen) do
 for each particle $i$ in the sub-swarm $k$ do
  for each particle $i$ do
   update velocity $v^k{}_i$ and position $x^k{}_i$ using Equations (6)–(11)
  end for
 calculate particle fitness value of $x^k{}_i$
  end for
if(generation $\geq$ R) randomly regrouped the neighbouring sub-swarms
 update individual best position $P_{best}$ and swarm best position $P_{gbest}$
 gen = gen + 1
end

**Figure 10.** MSPSO-IPTBK algorithm.

Input: Parameters of the MSPSO-IPTBK model, including the initial values of max_gen, $c_1$, $c_2$, $x$ and $v$ for the particles set, and the test data generated using the Waxman theory [24]
Output: Predicted accuracy of model

## 4. Results

This section discusses the applicability of the proposed MSPSO-TPBK model by presenting two botnet examples. For security concerns in academic networks, experiments were performed in simulated environment using ns-3 with the BRITE framework on a PC with a 3-GHz Intel Dual-Core CPU and 2 GB of DDR3 RAM running on Ubuntu Desktop 18.04.3 LTS. Network simulation for network security is a cost-effective method for evaluating, testing and selecting the required algorithm. In the designed experiments, defenders can examine all routing routes for DDoS attacks and evaluate the basic performance of IP trackback algorithms. The simulated experiments involved flow statistical analysis and packet marking using an SPIE and the creation of network topology using the ns-3 simulation tool with the BRITE framework. The ns-3 program is not an extension of ns-2. In practice, the software infrastructure of ns-3 allows it to be used as a real-time network emulator that can be connected with the real world. Numerous existing real-world protocol implementations can be reused within ns-3. Moreover, ns-3 also supports a real-time scheduler that facilitates numerous simulation-in-the-loop techniques [25].

The BRITE is a standard tool for generating realistic Internet topologies. The ns-3 mode provides a helper class to facilitate the generation of ns-3-specific topologies using BRITE configuration files. The BRITE constructs the original graph, which is stored in the form of nodes and edges in the ns-3 BriteTopolgyHelper class. The ns-3 integration of the BRITE generates a topology and then provides access to leaf nodes for each user. Thus,

ns-3 users can attach custom topologies to these leaf nodes by creating them manually or using the topology generators provided in ns-3 [26].

### 4.1. Case I: Network Performance Analysis for DDoS Attacks (24 Nodes)

The first example considers the profiles of DDoS attacks on IoT devices on a cloud server. A network intrusion detection system was constructed using the following three phases: (1) Data pre-processing, (2) attack path reconstruction and (3) model validation phases. The workflow of security analysis is illustrated in Figure 4.

#### 4.1.1. Step 1: Data Pre-Processing Phase

Creation of the Network Topology

The ns-3 software was deployed with BRITE to generate 24 nodes at integer coordinates over a rectangular area of 300 × 300, as displayed in Figure 11. As depicted in Figure 11, the simulated network topology consisted of two local area networks (LANs). Simulated hosts and routers were configured using a BriteTopologyHelper class. Moreover, each of the four LANs had host nodes, one switch node, one router node and the relay nodes of the internet. The attack sites were compromised IoT devices and were host0–host9 in LAN1. The switch node was designated as switch1. The victim was an online game server (host11) in LAN2 (host10–host19). We used the 'networkx,' which is a Python package to build the network topology. Each pair of adjacent nodes was an edge which was assigned a weight or cost in all paths. Then, we used the function attribute res_cost(x,y) to indicate the bandwidth and quality of service (QoS) of a path.

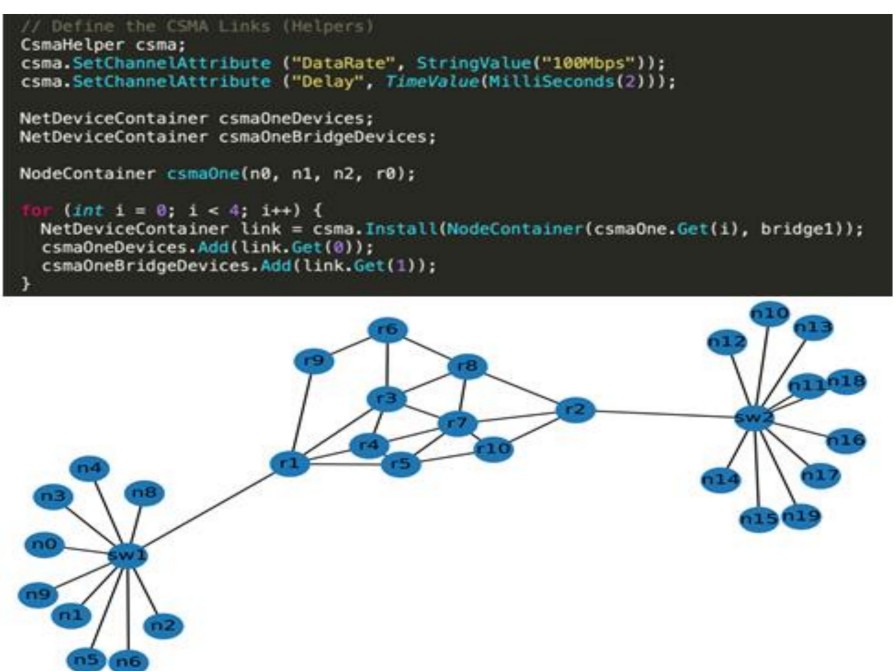

```
// Define the CSMA Links (Helpers)
CsmaHelper csma;
csma.SetChannelAttribute ("DataRate", StringValue("100Mbps"));
csma.SetChannelAttribute ("Delay", TimeValue(MilliSeconds(2)));

NetDeviceContainer csmaOneDevices;
NetDeviceContainer csmaOneBridgeDevices;

NodeContainer csmaOne(n0, n1, n2, r0);

for (int i = 0; i < 4; i++) {
  NetDeviceContainer link = csma.Install(NodeContainer(csmaOne.Get(i), bridge1));
  csmaOneDevices.Add(link.Get(0));
  csmaOneBridgeDevices.Add(link.Get(1));
}
```

**Figure 11.** Simulated network topology specified by the Boston University Representative Internet Topology Generator (BRITE) framework (number of nodes = 24).

One must set the routing cost of each route in the network topology to decide the next hop path using the command res_cost = array ([*x*, *y*], weight = *x*). The lower the routing cost is on a path, the higher priority a packet has on that path. A high-priority packet can traverse a low-cost path with a relatively small delay. In the experiments, equal cost (i.e., res_cost was set as 1) was adopted to examine the performance of the proposed mode.

Step 1.2: Data Pre-Processing for DDoS Threats

The attack paths were constructed using the following two-step procedure.

Step 1.2.1: Attack on the victim

In this step, the attack nodes host0–host2 (IP address of 192.158.1.2–192.168.1.4) launched a DDoS attack using UDP floods against the online game server (host11) in LAN2. The victim (IP address of 192.168.2.3) listened by default on port 5001. Six cycles of attacks were conducted in 120 s to generate routing information on the victim node for the IPTBK problem. A total of 1800 attack packets ($m = 1800$) were sent to host4 using UDP floods. The average packet quantity of the visited node was the basis of updating the number of particles and assisting particle swarms to trace the sources of attacks by reconstructing route routes, as illustrated in Figure 12.

```
08:00:01.003000 ARP, Request who-has 192.168.1.2 (Broadcast) tell 192.168.1.1, length 50
08:00:01.007011 ARP, Reply 192.168.1.2 is-at 00:00:00:00:00:02 (oui Ethernet), length 50
08:00:01.007011 IP 192.168.1.1.49153 > 192.168.5.1.terabase: UDP, length 1024
08:00:04.000000 IP 192.168.1.1.49153 > 192.168.5.1.terabase: UDP, length 1024
08:00:07.000000 IP 192.168.1.1.49153 > 192.168.5.1.terabase: UDP, length 1024
08:00:10.000000 IP 192.168.1.1.49153 > 192.168.5.1.terabase: UDP, length 1024
08:00:13.000000 IP 192.168.1.1.49153 > 192.168.5.1.terabase: UDP, length 1024
08:00:16.000000 IP 192.168.1.1.49153 > 192.168.5.1.terabase: UDP, length 1024
08:00:19.000000 IP 192.168.1.1.49153 > 192.168.5.1.terabase: UDP, length 1024
08:00:22.000000 IP 192.168.1.1.49153 > 192.168.5.1.terabase: UDP, length 1024
08:00:25.000000 IP 192.168.1.1.49153 > 192.168.5.1.terabase: UDP, length 1024
08:00:28.000000 IP 192.168.1.1.49153 > 192.168.5.1.terabase: UDP, length 1024
```

**Figure 12.** Routing routes for Distributed Denial-Of-Service (DDoS) attacks from host0–host2.

Step 1.2.2: Data collection

In the experiments, we used Wireshark to collect samples of network traffic flows on port 5001 of the victim. The traffic flows were recorded in the Pcap format. Once the attack flow packets had been collected, a set of tools (scavetool) were applied to convert the recording files to the comma-separated values (CSV) format for the reconstruction of attack paths, as depicted in Figure 13.

```
                            run       type module name  ... min max  binedges  binvalues
0  dynamic2-0-20191027-09:09:54-1699  runattr  NaN  NaN  ... NaN NaN      NaN        NaN
1  dynamic2-0-20191027-09:09:54-1699  runattr  NaN  NaN  ... NaN NaN      NaN        NaN
2  dynamic2-0-20191027-09:09:54-1699  runattr  NaN  NaN  ... NaN NaN      NaN        NaN
3  dynamic2-0-20191027-09:09:54-1699  runattr  NaN  NaN  ... NaN NaN      NaN        NaN
4  dynamic2-0-20191027-09:09:54-1699  runattr  NaN  NaN  ... NaN NaN      NaN        NaN
```

**Figure 13.** Conversion of network packet files to the CSV format.

### 4.1.2. Step 2: Route Construction Phase

The routing information generated in Step 1.2.2 was used as the input dataset of the PSO model. The important characteristics of the PSO model were as follows: (1) The population of the particle was set to the number of packets collected for DDoS attacks; (2) the loop executed 100 generations and updated route-searching rules for each loop; (3) the initial value of $w_i$ (weighting factor) was 0.8; (4) $c_1$ was equal to $c_2$, and they were set as 2.0 in Equation (4); and (5) the number of subswarms was 4.

**Scenario 1:** The routing costs of all routes are the same (res_cost = 1).

In the experiments, res_cost = numpy.array ([$x$, $y$]) was used to set the routing cost for paths $r1$–$r5$–$r7$–$r2$ and $r1$–$r5$–$r10$–$r2$ (Figure 13). Figure 14 indicates that all paths had the same routing cost and weighting. The higher the routing cost, the lower was the routing priority. The larger the weight $w$, the larger was the bandwidth. The shorter the routing distance for a path, the higher was the QoS on that path.

```
# Add Edges
G.add_edges_from([("r1", "r5"), ("r1", "r4"), ("r1", "r3"), ("r1", "r9")],
                 res_cost=array([1, 1]), weight=1)
G.add_edges_from([("r5", "r4")], res_cost=array([1, 1]), weight=1)
G.add_edges_from([("r5", "r7")], res_cost=array([1, 1]), weight=1)
G.add_edges_from([("r5", "r10")], res_cost=array([1, 1]), weight=1)
G.add_edges_from([("r7", "r2")], res_cost=array([1, 1]), weight=1)
G.add_edges_from([("r10", "r2")], res_cost=array([1, 1]), weight=1)
```

**Figure 14.** Static routing cost settings for paths $r1$–$r5$–$r7$–$r2$ and $r1$–$r5$–$r10$–$r2$.

The system had 118 routes between $n_0$ (the attack node) and $n_{11}$ (the victim). This information was obtained using the all_simple_paths API call in the *network* component of ns-3, which is described in Table 3.

**Table 3.** Set of 118 possible routes between the attack node ($n_0$) and the victim ($n_{11}$).

| |
|---|
| $n_0$-Switch1-router1-router5-router4-router3-router8-router7-router10-router2-Switch2-$n_{11}$ |
| $n_0$-Switch 1-router1-router5-router4-router3-router8-router7-router2- Switch2-$n_{11}$ |
| $n_0$-Switch 1-router1-router5-router4-router3-router8- router2- Switch2-$n_{11}$ |
| $n_0$-Switch 1-router1-router9-router6-router8-router2- Switch 2-$n_{11}$ |

Using the A* search algorithm

The A* algorithm [19] is an improved Dijkstra's algorithm. It can be used to identify the shortest path between any two end nodes in the search space. This study compared the performance of both the A* and MSPSO algorithms for solving the IPTBK problem.

First, we used the A* (networkx suite) algorithm for the attack nodes $n_0$, $n_1$ and $n_2$ attacking $n_{11}$. The smallest cost of the attack path in this case is as follows:

Route 1: ['$n_0$', '$sw1$', '$r1$', '$r5$', '$r7$', '$r2$', '$sw2$', '$n_{11}$'],
Route 2: ['$n_1$', '$sw1$', '$r1$', '$r5$', '$r7$', '$r2$', '$sw2$', '$n_{11}$'],
Route 3: ['$n_2$', '$sw1$', '$r1$', '$r5$', '$r7$', '$r2$', '$sw2$', '$n_{11}$']

Using the MSPSO algorithm

We used the MSPSO algorithm to analyse the possible attack path for the test case ($n_0 \rightarrow n_{11}$).

Through the reconstruction of the attack path for the attack case ($n_0 \rightarrow n_{11}$), we applied the subgroup searching strategy to obtain the optimal solution using the MSPSO-TPBK algorithm. Particles travelled around all the paths and back to the attack origins according to the local and global updating rules presented in Equations (6)–(11). After 100 generations had been executed, the results revealed three possible attack paths for conducting model performance analysis.

Route 1: ['$n_0$', '$sw1$', '$r1$', '$r5$', '$r7$', '$r2$', '$sw2$', '$n_{11}$']
Route 2: ['$n_0$', '$sw1$', '$r1$', '$r4$', '$r7$', '$r2$', '$sw2$', '$n_{11}$']
Route 3: ['$n_0$', '$sw1$', '$r1$', '$r3$', '$r8$', '$r2$', '$sw2$', '$n_{11}$']

4.1.3. Step 3: Model Validation Phase

Using Equation (13), the coverage rate of the attack path in the experimental case was calculated (Table 4). In Table 4, the first three columns were selected as possible attack paths for the experimental case, where the minimum support threshold was $t = 3\%$. Table 4 indicates that the MSPSO-IPTBK accuracy considering that the static traffic was 95.83% and that the error rate was 4.17% for the network topology (number of nodes = 32).

**Table 4.** Possible paths of DDoS attacks (number of nodes = 32, array ($x$, $y$), $w = 1$).

| Attack Path | Packets Collected | Coverage Percentage (%) |
|---|---|---|
| $n_0$-switch1-router1-router5-router7-router2-switch2-$n_{11}$ | 500 | 41.67% |
| $n_0$-switch1-router1-router4-router7-router2-switch2-$n_{11}$ | 500 | 41.67% |
| $n_0$-switch1-router1-router3-router8-router2-switch2-$n_{11}$ | 170 | 14.17% |
| Total | 1200 | 100.0% |

**Scenario 2:** Dynamic routing cost of routes

Considering the factors of traffic dynamics, including the bandwidth, traffic delay and QoS requirements, we set the different weights (i.e., cost) of each route in the network

topology to decide the next hop path using the command res_cost = array ($[x, y]$, $w = x$) for paths $r1$–$r5$–$r7$–$r2$ and $r1$–$r5$–$r10$–$r2$ (Figure 15).

```
# Add Edges
G.add_edges_from([("r1", "r5"), ("r1", "r4"), ("r1", "r3"), ("r1", "r9")],
                 res_cost=array([10, 8]), weight=8)
G.add_edges_from([("r5", "r4")], res_cost=array([1, 1]), weight=1)
G.add_edges_from([("r5", "r7")], res_cost=array([8, 6]), weight=6)
G.add_edges_from([("r5", "r10")], res_cost=array([5, 5]), weight=5)
G.add_edges_from([("r7", "r2")], res_cost=array([10, 8]), weight=8)
G.add_edges_from([("r10", "r2")], res_cost=array([5, 5]), weight=5)
```

**Figure 15.** Setting of different weights for each route in the network topology.

Using the A* search algorithm

The same results were obtained as in the previous case.

Using the MSPSO algorithm

We used the MSPSO algorithm to analyse the possible attack paths for the test case ($n_0 \rightarrow n_{11}$).

The static and dynamic routing cases were examined as described in the following text.

After 100 generations, three possible attack paths were identified for the static routing case (we set $w = 1$ for each path).

Route 1: ['$n_0$', 'sw1', 'r1', 'r5', 'r7', 'r2', 'sw2', '$n_{11}$']
Route 2: ['$n_0$', 'sw1', 'r1', 'r4', 'r7', 'r2', 'sw2', '$n_{11}$']
Route 3: ['$n_0$', 'sw1', 'r1', 'r3', 'r8', 'r2', 'sw2', '$n_{11}$']

Similarly, four possible attack paths were identified for the dynamic routing case ($w = x$), where $x$ represents a variable that depends on the bandwidth, QoS and traffic delay.

Route 1: ['$n_0$', 'sw1', 'r1', 'r5', 'r7', 'r2', 'sw2', '$n_{11}$']
Route 2: ['$n_0$', 'sw1', 'r1', 'r4', 'r7', 'r2', 'sw2', '$n_{11}$']
Route 3: ['$n_0$', 'sw1', 'r1', 'r3', 'r8', 'r2', 'sw2', '$n_{11}$']
Route 4: ['$n_0$', 'sw1', 'r1', 'r9', 'r3', 'r8', 'r2', 'sw2', '$n_{11}$']

For the dynamic routing case ($w = x$), the coverage rates of the attack paths are listed in Table 5. Table 5 indicates that the accuracy of the MSPSO algorithm is 97.50%, and the false rate is 2.5% when the number of network topology nodes is 32. The experimental results of attack path reconstruction when considering traffic dynamics (i.e., $w = x$) are superior to those obtained when the effects of traffic dynamics were not considered (i.e., $w = 1$ for each path). Thus, the proposed MSPSO-IPTBK model is flexible for different routing applications.

**Table 5.** Possible paths of DDoS attacks (number of nodes = 32, array ($x, y$), $w = x$).

| Attack Path | Packets Collected | Coverage Percentage (%) |
|---|---|---|
| $n_0$-switch1-router1-router5-router7-router2-switch2-$n_{11}$ | 500 | 41.67% |
| $n_0$-switch1-router1-router4-router7-router2-switch2-$n_{11}$ | 500 | 41.67% |
| $n_0$-switch1-router1-router3-router8-router2-switch2-$n_{11}$ | 150 | 12.50% |
| $n_0$-switch1-router1-router9-router3-router8-router2-switch2-$n_{11}$ | 20 | 1.67% |
| Total | 1200 | 100.0% |

### 4.2. Case II: Performance Analysis for a Series of DDoS Attacks (64 Nodes)

The second example considered the effect of the network size on the number of packets required to construct the attack path. For similar test runs, a series of DDoS attacks were conducted in a simulated network topology (number of nodes = 64) to evaluate the convergence performance of the proposed model, as illustrated in Figure 16.

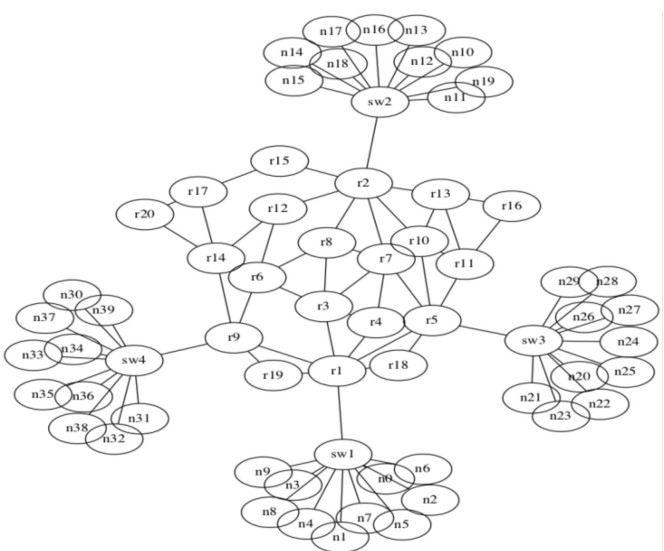

**Figure 16.** Simulated network topology (number of nodes = 64).

In the experiments, four LANs (i.e., Lan1 to Lan4) were generated by the BRITE, as displayed in Figure 17. The attacker flooded the victim ($n_{11}$) with packets originating from three groups of attack nodes ($n_0$, $n_1$ and $n_2$; $n_{20}$, $n_{21}$ and $n_{22}$; and $n_{30}$, $n_{31}$ and $n_{32}$) in LAN1, LAN3 and LAN4. A total of 1200 attack packets were sent in irregular bursts to congest the link in 30 s.

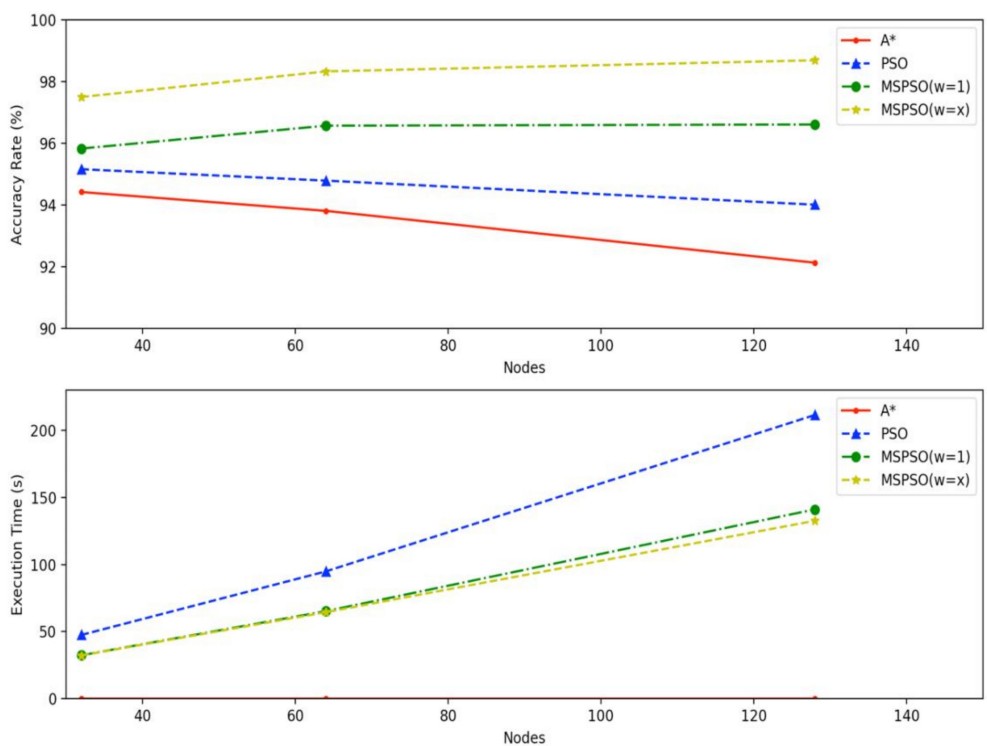

**Figure 17.** Execution time of route reconstruction for IP traceback using three path-search algorithms.

Using the A* search algorithm

We used the A* algorithm with 'networkx' software package for the attack node $n_0$ attacking $n_{11}$ with nine attacks. In this case, the smallest cost of the attack path can be expressed as follows:

Route 1: ['$n_0$', '$sw1$', '$r1$', '$r5$', '$r7$', '$r2$', '$sw2$', '$n_{11}$']

Route 2: ['$n_1$', '$sw1$', '$r1$', '$r5$', '$r7$', '$r2$', '$sw2$', '$n_{11}$']
Route 3: ['$n_2$', '$sw1$', '$r1$', '$r5$', '$r7$', '$r2$', '$sw2$', '$n_{11}$']
Route 4: ['$n_{20}$', '$sw3$', '$r5$', '$r7$', '$r2$', '$sw2$', '$n_{11}$']
Route 5: ['$n_{21}$', '$sw3$', '$r5$', '$r7$', '$r2$', '$sw2$', '$n_{11}$']
Route 6: ['$n_{22}$', '$sw3$', '$r5$', '$r7$', '$r2$', '$sw2$', '$n_{11}$']
Route 7: ['$n_{30}$', '$sw4$', '$r9$', '$r6$', '$r8$', '$r2$', '$sw2$', '$n_{11}$']
Route 8: ['$n_{31}$', '$sw4$', '$r9$', '$r6$', '$r8$', '$r2$', '$sw2$', '$n_{11}$']
Route 9: ['$n_{32}$', '$sw4$', '$r9$', '$r6$', '$r8$', '$r2$', '$sw2$', '$n_{11}$']

Using the MSPSO algorithm

The MSPSO algorithm was applied to obtain the following four possible attack paths for the $n_0 \rightarrow n_{11}$ attack case:

Route 1: ['$n_0$', '$sw1$', '$r1$', '$r5$', '$r7$', '$r2$', '$sw2$', '$n_{11}$']
Route 2: ['$n_0$', '$sw1$', '$r1$', '$r4$', '$r7$', '$r2$', '$sw2$', '$n_{11}$']
Route 3: ['$n_0$', '$sw1$', '$r1$', '$r3$', '$r8$', '$r2$', '$sw2$', '$n_{11}$']
Route 4: ['$n_0$', '$sw1$', '$r1$', '$r9$', '$r3$', '$r8$', '$r2$', '$sw2$', '$n_{11}$']

Similarly, the MSPSO was applied to obtain the following three possible attack paths for the attack case $n_{20} \rightarrow n_{11}$:

Route 1: ['$n_{20}$', '$sw3$', '$r5$', '$r7$', '$r2$', '$sw2$', '$n_{11}$']
Route 2: ['$n_{20}$', '$sw3$', '$r5$', '$r4$', '$r7$', '$r2$', '$sw2$', '$n_{11}$']
Route 3: ['$n_{20}$', '$sw3$', '$r5$', '$r11$', '$r10$', '$r2$', '$sw2$', '$n_{11}$']

Moreover, the following two possible attack paths were obtained for the attack case $n_{30} \rightarrow n_{11}$:

Route 1: ['$n_{30}$', '$sw4$', '$r9$', '$r3$', '$r8$', '$r2$', '$sw2$', '$n_{11}$']
Route 2: ['$n_{30}$', '$sw4$', '$r9$', '$r6$', '$r8$', '$r2$', '$sw2$', '$n_{11}$']

Subsequently, we applied the subgroup searching strategy to obtain the optimal solution using four subswarms of the MSPSO-TPBK algorithm. The coverage percentages for the attack cases $n_0 \rightarrow n_{11}$, $n_{20} \rightarrow n_{11}$ and $n_{30} \rightarrow n_{11}$ are listed in Table 4.

Table 6 indicates that the MSPSO-IPTBK accuracy was 98.33% for the network topology (number of nodes = 64), and the error rate was 1.67%. Similarly, experiments were conducted by setting different weights on each route of the network topology. In contrast to the results in Tables 4 and 5, the experimental results in Table 6 indicate that a large network preserves the convergence performance of the MSPSO-IPBK scheme.

**Table 6.** Possible attack path of DDoS attacks (number of nodes = 64, array $(x, y)$, $w = x^*$).

| Attack Path | Packets Collected | Coverage Percentage (%) |
|---|---|---|
| $n_0$-switch1-router1-router5-router7-router2-switch2-n$_{11}$ | 200 | 16.67% |
| $n_0$-switch1-router1-router4-router7-router2-switch2-n$_{11}$ | 200 | 16.67% |
| $n_0$-switch1-router1-router3-router8-router2-switch2-n$_{11}$ | 20 | 1.67% |
| $n_0$-switch1-router1-router9- router3-router8-router2-switch2-n$_{11}$ | 20 | 1.67% |
| $n_{20}$-switch3- router5-router7-router2-switch2-n$_{11}$ | 280 | 23.33% |
| $n_{20}$-switch3-router5-router4-router7-router2-switch2-n$_{11}$ | 280 | 23.33% |
| $n_{20}$-switch3-router5-router11-router10-router2-switch2-n$_{11}$ | 140 | 11.67% |
| $n_{30}$-switch4-router9-router3-router8-router2-switch2-n$_{11}$ | 20 | 1.67% |
| $n_{30}$-switch4-router9-router6-router8-router2-switch2-n$_{11}$ | 20 | 1.67% |
| Total | 4071 | 100.0% |

* $x$ is a variable that depends on the bandwidth, quality of service (QoS) and traffic delay.

To evaluate the effectiveness of the proposed MSPSO algorithm in defending DDoS attacks, different sizes of network topologies were tested. In experiment II, BRITE generated four LANs (i.e., Lan1 to Lan4). The attacker was alternatively flooding the victim ($n_{11}$) with packets originating from the attack node groups $n_0$, $n_1$ and $n_2$; $n_{20}$, $n_{21}$ and $n_{22}$; and

$n_{30}$, $n_{31}$ and $n_{32}$ in LAN1, LAN3 and LAN4, respectively. The sizing of the node set was varied to examine the traceback accuracy according to the coverage rate (%) of particles on the attack path. Three path-search algorithms were employed to examine the performance analysis for DDoS attacks. The experimental parameter settings for PSO algorithm were num_particles = 1200 and epochs = 100 and num_swarms = 4, num_particles = 300 and epochs = 100 for the MSPSO algorithm. The experimental results are presented in Table 7 and Figure 17. Table 7 lists the accuracy and execution time for a test set of routing algorithms with different topology sizes. The experimental results indicate that the traceback error decreased as the size of the testing data increased. The prediction accuracy increased with an increase in the size of the network topology (*n*). The overall accuracy rate for the three test cases was 98.17%. Figure 17 shows that the execution time of the proposed algorithm for dynamic routing (*w* = *x*) is higher than those of the A* and the PSO algorithm because the regrouping strategy needs a higher computational time, but it extends the search space of the global optimal solution.

**Table 7.** Traceback accuracy vs. execution time of DDoS attacks when using the proposed algorithm with two path-search algorithms.

| Scheme / Topology | *n* = 32 Nodes | *n* = 64 Nodes | *n* = 128 Nodes |
|---|---|---|---|
| A* search algorithm | 94.42%/0.95 msec | 93.81%/4.64 msec | 92.13%/9.98 msec |
| PSO | 95.16%/47,482.19 msec | 94.79%/396.17 msec | 94.01%/211,453.41 msec |
| MSPSO (*w* = 1) | 95.83%/32,125.30 msec | 96.57%/65,209.73 msec | 96.61%/141,071.80 msec |
| MSPSO (*w* = *x*) | 97.50%/32,202.44 msec | 98.33%/64,434.63 msec | 98.69%/132,566.52 msec |

## 5. Conclusions

This study presents an MSPSO-based IPTBK model that can analyse the effects of multiple-swarm search strategies under different topology sizes and optimal fitness function settings for the quality of PSO solutions to enhance the reconstruction of the probable paths of DDoS attacks. The experimental results confirm that the proposed method can analyse the possible attack paths of botnets and the attack sources of DDoS threats.

Although MSPSO techniques have been proposed for the IPTBK problem, there are practical challenges of using the proposed method. For example, in case of a DDoS attack, the proposed method generates unusually high rates of packet loss across a network that may disorder the sequence of packet marking, and therefore may harmfully affect the traceback accuracy of MSPSO. Moreover, it needs to integrate the proposed scheme into the traceback module of intrusion detection system (IDS). The scalability challenge and convergence of large complex network topologies (number of nodes > 1024) in the IPTBK problem will be addressed in a future study. Behavioural analysis with classification rules may be considered for determining the class of new threat in the future to implement rapid countermeasures against cyber-attacks.

**Author Contributions:** Conceptualization, P.W.; methodology, H.-C.L.; resources, P.W.; formal analysis, H.-C.L.; data curation, H.-C.L.; writing—original draft, H.-C.L. and W.-H.L.; writing-review and editing, P.W.; software, H.-C.L. and Y.-H.H.; validation, H.-C.L. and W.-H.L.; visualization, H.-C.L. and Y.-H.H.; project administration, P.W.; funding acquisition, P.W. All authors have read and agreed to the published version of the manuscript.

**Funding:** This research was funded by the Ministry of Science and Technology of Taiwan under Grant Nos. MOST 108-3116-F-168-001-CC2, and MOST 109-2410–H-168-005.

**Institutional Review Board Statement:** Not applicable.

**Informed Consent Statement:** Informed consent was obtained from all subjects involved in the study.

**Data Availability Statement:** The data presented in this study are available on request from the corresponding author.

**Acknowledgments:** This work was supported jointly by the Ministry of Science and Technology of Taiwan under Grant Nos. MOST 108-3116-F-168-001-CC2, and MOST 109-2410–H-168-005.

**Conflicts of Interest:** The authors declare no conflict of interest.

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
