# Peer review of "A Multiple-Swarm Particle Swarm Optimisation Scheme for Tracing Packets Back to the Attack Sources of Botnet†"

_applsci, doi:10.3390/app11031139_

Round 1

Reviewer 1 Report

This paper described a multiple-swarm particle swarm optimization (MSPSO) scheme for tracing packets back to the attack sources of botnet, and experiments showed that the MSPSO algorithm performs better in multimodal optimization problems, improves the accuracy of traceability analysis.

Overall, the paper is well-organized and easy to follow.

I like the following part of the paper:

1) clear explanation about the MSPSO algorithm;

2) a detailed description of the subphases in the attack path reconstruction process;

3) accuracy results comparison between different approaches.

One potential improvement I can think of is to also compare the efficiency between different approaches in the evaluation.

Author Response

Q1. This paper described a multiple-swarm particle swarm optimization (MSPSO) scheme for tracing packets back to the attack sources of botnet, and experiments showed that the MSPSO algorithm performs better in multimodal optimization problems, improves the accuracy of traceability analysis. Overall, the paper is well-organized and easy to follow.

Plesae clear explanation about the MSPSO algorithm.

Ans: Thanks for valuable comments. The background and definition of MSPSO algorithm is addressed in Line 113-118 of Section 1 (in brown colour).  

-------------------------------------------------------------------------------------------------------------------

Q2. A detailed description of the subphases in the attack path reconstruction process.

Ans: The detailed description of the subphases in the attack path reconstruction process is addressed in Section 3.2.2. Route Construction Phase. There are two sub-pahses in this process, i.e., 3.2.2.1 Local search process and 3.2.2.2 Global search process.

     Also the supplement of route construction is added in Line 340-342 (in brown colour)

-----------------------------------------------------------------------------------------------------------

Q3. The accuracy results comparison between different approaches..

Ans: Thanks for valuable comments.

This study compared the performance of both the A* and MSPSO algorithms for solving the IPTBK problem. In Table 7, we compared the experimental accuracy with two similar route search algorithms (i.e., traditional PSO and A* search algorithm) to validate the trackback accuracy in reconstruction of attacl paths. The A* search algorithm is an improved Dijkstra's algorithm. It can be used to identify the shortest path between any two end nodes in the search space.

-----------------------------------------------------------------------------------------------------------

Q4. One potential improvement I can think of is to also compare the efficiency between different approaches in the evaluation.

Ans: We add the the efficiency comparison with execution time for three path-search algorithms (i.e., A* algorithm, traditional PSO, MSPSO with dynamic routing) in the Table 7 and Figure.17 to improve credibility.

Reviewer 2 Report

The article is considered relevant from a technical point of view, however, it is considered that it must improve in the following aspects:

  • Abstract: distributed denial-23 of-service (DDoS) to put the initial letter of each word in upper case
  • To improve the following paragraph:
    • 37 Hackers then perform cyber attacks on the
    • 38 compromised computers. Attacks include terminating the web services of opponents through
    • 39 distributed denial-of-service (DDoS) attacks and theft of business information
  • There are more types of attacks, not only these two types, why is it included if you are talking in a general way about one so specific to web services?
  • The following paragraph is considered to refer to figure 1 not figure 2. “60 Two main types of bots are used in DDoS attacks: IRC and peer-to-peer bots. As illustrated in Figure 2, attack path reconstruction is a typical 61 IPTBK technique used against IRC bot attacks. As displayed in Figure 2, attackers use zombies”
  • To finalize the sentence “113 The validity of the proposed algorithm was demonstrated by performing a series of experiments by using ns-3 “ by the word simulator, program, etc..
  • According to the journal all figures and tables should be cited in the main text as Figure 1, Table 1, etc. For example, in line 192 it is included the cite (Fig.3) no according to the rules of the journal.
  • To improve the paper it is recommended to include a table, similar to table 1, that compare all the algorithm of IPTBK included in it.
  • The simulator used to simulate the network may not be representative of a real network such as the Internet. Please justify in more detail the validity of your simulated network and include its limitation.
  • The conclusions of the paper need to be improved by making it clearer and in major detail what are its main contributions.

Author Response

-Reviewer 2

The article is considered relevant from a technical point of view, however, it is considered that it must improve in the following aspects:

Thank You for valuable comments and suggestions. The comments and suggestions are valuable and very helpful for revising and improving our manuscript.

Q1 Abstract: distributed denial- of-service (DDoS) to put the initial letter of each word in upper case

Ans: Thanks for valuable comments. Revised the wordiness of distributed denial- of-service (DDoS) in abstract by using the initial letter of each word in upper case

-------------------------------------------------------------------------------------------------------------------

Q2 To improve the following paragraph: Hackers then perform cyber attacks on the compromised computers. Attacks include terminating the web services of opponents through distributed denial-of-service (DDoS) attacks and theft of business information. There are more types of attacks, not only these two types, why is it included if you are talking in a general way about one so specific to web services?

Ans: Thanks for valuable comments. To improve recognition of cyber attacks, the common types of cybersecurity attacks are included. Also the threat specific to web services from DDoS attacks are suppled and the description of this paragraph was rewritten. (Line 38-40)

-----------------------------------------------------------------------------------------------------------

Q3 The following paragraph is considered to refer to figure 1 not figure 2. “Two main types of bots are used in DDoS attacks: IRC and peer-to-peer bots. As illustrated in Figure 2, attack path reconstruction is a typical IPTBK technique used against IRC bot attacks. As displayed in Figure 2, attackers use zombies”

Ans: Thanks for valuable comments. Correct the figure 2 to figure 1.

-----------------------------------------------------------------------------------------------------------

Q4 To finalize the sentence “113 The validity of the proposed algorithm was demonstrated by performing a series of experiments by using ns-3 “ by the word simulator, program, etc..

Ans: Thanks for valuable comments. We revised this paragraph for clarity. (Line 124-125)

-----------------------------------------------------------------------------------------------------------

Q5. According to the journal all figures and tables should be cited in the main text as Figure 1, Table 1, etc. For example, in line 192 it is included the cite (Fig.3) no according to the rules of the journal.

Ans: Thanks for valuable comments. We add the citation to Fig.3 based on the original reference.

-----------------------------------------------------------------------------------------------------------

Q6 To improve the paper it is recommended to include a table, similar to table 1, that compare all the algorithm of IPTBK included in it.

Ans: Thanks for valuable comments. We summarized IP traceback techniques of IPTBK in  Table 1.

-----------------------------------------------------------------------------------------------------------

Q7. The simulator used to simulate the network may not be representative of a real network such as the Internet. Please justify in more detail the validity of your simulated network and include its limitation.

Ans: Thanks for valuable comments. Add why select the simulated networks to examine the routing routes and trackback results in DDoS attacks (Line 406-411) and details about simulated network (Line 437-440, 602-605) and the limitations of the simulated model in the conclusion. (Line 625-629)

-----------------------------------------------------------------------------------------------------------

Q8. The conclusions of the paper need to be improved by making it clearer and in major detail

what are its main contributions.

Ans: To point out the technical contribution of the paper, we summarize the technical achievements of this work and also the features of proposed approach are revised in the Introduction section. (Line 129-141)

Round 2

Reviewer 2 Report

From my point of view the paper has improved and I have no more comments as those made in the first revision have been resolved.
